# The Second Wave of COVID-19 Pandemic Strikes during the Flu Season: An Awareness Perspective

**DOI:** 10.3390/medicina56120707

**Published:** 2020-12-18

**Authors:** Alexandru Burlacu, Radu Crisan-Dabija, Iolanda Valentina Popa, Adrian Covic

**Affiliations:** 1Department of Interventional Cardiology—Cardiovascular Diseases Institute, 700503 Iasi, Romania; alburlacu@yahoo.com; 2Department of Internal Medicine, ‘Grigore T. Popa’ University of Medicine and Pharmacy, 700115 Iasi, Romania; crisanradu@gmail.com (R.C.-D.); accovic@gmail.com (A.C.); 3Pulmonology Department, Clinic of Pulmonary Diseases, 700115 Iasi, Romania; 4Nephrology Clinic, Dialysis, and Renal Transplant Center—‘C.I. Parhon’ University Hospital, 700503 Iasi, Romania

**Keywords:** COVID-19, SARS-CoV-2, flu season, influenza virus, coinfections

## Abstract

Coinfection with both SARS-CoV-2 and influenza viruses seems to be a real and severe problem. However, coinfection is far from a simple matter, and cannot be considered having more unfavorable outcomes as a direct consequence. In reality, the aftermath is powerfully nuanced by the presence of risk factors and specific molecular mechanisms. Our objective was to raise awareness around the unpredictable association between COVID-19 pandemics and the upcoming flu season, and make arguments about the need to develop new routine testing protocols for both viruses, at least during the period with an expected high incidence. Our reasoning is built around the various impacts that the whole range of risk groups, common immunological mechanisms, and complex interactions, such as influenza vaccination, will have on patients’ prognosis. We show that the more flawed clinical course is due to managing only one of the infections (and, subsequently, neglecting the other condition).

## 1. Introduction

“Winter is coming” and with it comes the flu season. In most countries, the second wave of COVID-19 is manifesting. Not only do both viruses arrive at once, but they can come together. Coinfection with both respiratory viruses seems to be a real and severe problem. Drawing two non-winning lotteries seems hard to believe, but several complex evolution cases are already described [1].

The issue of coinfection is far from a simple matter and cannot be considered to have more unfavorable outcomes. In reality, the aftermath is powerfully nuanced by a whole range of risk groups, common immunological mechanisms, and complex interactions, such as influenza vaccination, with an undeniable impact on patients’ prognosis.

Viral coinfection seems not to have a linear worsening; therefore, several questions remain to be answered: Which of the viruses poses a greater risk of death? Wherewith the flu vaccine influences COVID-19 infection? What common mechanisms can be targeted by (old and novel) therapies, as failing to diagnose one of the infections may also lead to inappropriate treatment?

## 2. Is There a Specific Category of Patients Prone to Viral Coinfection?

The answer to this question is far from clear. However, there are some categories with specific risk for unfavorable disease course. The importance of correctly identifying risk groups derives from the need to initiate appropriate treatment in a timely manner, as early treatment may lessen the risk of hospitalization and death [2].

Firstly, a retrospective study of the 2009 H1N1pdm influenza pandemic showed that a pre-existing chronic condition (older age [3], obesity, Type 2 diabetes mellitus, respiratory or cardiovascular diseases [4]) increased the likelihood of ICU admittance and fatal outcomes [5]. Hypertension, other cardiovascular diseases, and older age seem to be critical risk factors for both flu and COVID-19 severity, as was shown in a recent meta-analysis [6]. Obesity appears to be one of the most critical risk factors for both critical forms of H1N1 infection and severe COVID-19 [7]. A disruptive effect of the adipocytes [8] is at the front of disturbances in inflammatory crosstalk between leukocyte-mediated processes and molecular signaling [9], and alterations in leukocyte population and lymphocyte activity, leading to an insufficient immunological response [9,10] and predisposing to respiratory infections (particularly to severe forms of both COVID-19 [11] and influenza [12]).

Secondly, physical inactivity and insulin resistance interfere with macrophage activation and inhibit pro-inflammatory cytokines, predisposing obese patients to a double risk of contracting both viruses [13]. Type 2 diabetes quadruples the risk of hospitalization with severe forms of H1N1p infection [14,15]. Since the angiotensin-converting enzyme 2 (ACE2) receptor expression is significantly higher in Type 1 and Type 2 diabetes [16], these patients are at a double infection risk for SARS-CoV-2, according to several studies [6,17].

Thirdly, chronically ill respiratory patients (particularly chronic obstructive pulmonary disease [18] or tuberculosis sequelae) risk contracting both influenza and SARS-CoV-2 [19]. Active or past tuberculosis has an increased risk of severe outcomes during influenza infection [20,21] and a significantly higher risk for severe SARS-Cov-2 infection [22].

Fourthly, pregnant women were identified as being at increased risk for hospitalization and severe disease during the 2009 H1N1pdm influenza pandemic [5], while pregnant and recently pregnant women with COVID-19 were shown to be at increased risk of admission to an intensive care unit, according to the results of a recent meta-analysis [23].

Fifthly, oncologic patients under active treatment infected with SARS-CoV-2 were found to have worse outcomes in terms of mortality and respiratory failure rates, compared with COVID-19 in the general population [24]. Similarly, patients diagnosed with cancer or individuals under immunosuppressant medication were significantly associated with fatal outcomes when infected with influenza [25,26]. However, for immunocompromised individuals with COVID-19, “current studies have not shown worse outcomes, except for patients with cancer” [27].

Any of these patients should be carefully considered at risk for coinfection when respiratory symptoms are present.

## 3. Is One Plus One Equal to Two? The Clinical Outcomes of Patients Infected with Both SARS-CoV-2 and Influenza Viruses

Once one [1] has noted the increasing incidence of the two viruses’ coexistence, a legitimate question arises: does the presence of the two in the same patient determine a more severe evolution than each taken separately? A fair answer is relevant, as both viruses mimic each other, regarding clinical presentation, transmission, and seasonal coincidence. This important overlap may mask possible coinfections and may lead to inappropriate treatment strategies when clinical outcomes of viral coexistence are not known.

In other words: does one plus one equal two, or 0.5, or 4? To date, there is no observational study to assess coinfected patients’ evolution and outcomes compared to those infected with SARS-CoV-2 or other distinct respiratory viruses. What is clear is that an erroneous diagnostic, treatment, or both (either for COVID-19 or for the flu) can allow the clinical course of the other coinfectant to be “natural” (more severe) [28].

Several small case series have been published to date. Two small case series studies published in China reported, on the one hand, that “coinfection patients did not appear to experience a more difficult situation,” and, on the other hand, that “we cannot ignore COVID-19 infection patients might combine with other respiratory viruses” [29,30,31]. Another two case series in which almost all patients had a medical history of hypertension, and a significant proportion of them were diabetic or on hemodialysis, were published. Although almost all patients were classified as being part of high-risk categories, their clinical evolution did not differ from previously reported monoinfections with SARS-CoV-2 [32,33]. However, severe or fatal COVID-19 outcomes have also been described in isolated case reports of coinfections with influenza [34] or other viral, bacterial [35], or fungal pathogens (especially Aspergillus fumigatus) [36].

Another theory is that one of the respiratory viruses can actively help a viral strain (e.g., SARS-CoV-2) to invade, and thus, patients who otherwise would not have had COVID-19 would become infected with a significant viral load [37]. On the other hand, a study reporting on coinfection rates between SARS-CoV-2 and other respiratory viral and bacterial pathogens showed “no significant difference in rates of SARS-CoV-2 infection in patients with and without other pathogens” [38].

Moreover, a mathematical model of coinfections indicates that viruses with a faster growth rate will suppress viruses with a slower growth rate, in a viral competition for resources [39]. SARS-CoV-2 has a slower growth rate than influenza, indicating that SARS-CoV-2 replication may be suppressed when the novel coronavirus infection is initiated simultaneously or after the influenza infection [40]. When SARS-CoV-2 is the first to meet the human body, the suppression may be mitigated, to a limited extent [40]. If the suppression theory proves to be right, the therapeutic conduct should be adapted accordingly, since treating the influenza infection might warrant greater susceptibility for SARS-CoV-2. This emphasizes, even more, the importance of testing all symptomatic patients for both viruses, to make informed therapeutic decisions.

## 4. The Immunological Mechanisms of Influenza Infection Predisposing to SARS-CoV-2 Coinfection and Severity

In order to deepen the knowledge on viral coexistence beyond clinical symptoms, the study of molecular mechanisms may be able to explain already-known manifestations and may anticipate unrevealed clinical outcomes. A significant determinant of coinfection dynamics is the immune response by regulating viral entry sites and upregulating or downregulating different inflammatory pathways and receptors involved in the overall response to infection.

A critical immune response of the COVID-19—flu interaction might be the toll-like receptor 4 (TLR4). COVID-19 patients upregulate TLR4 pro-inflammatory signaling, as TLR4 is most likely responsible for recognizing molecular patterns from SARS-CoV-2 [41]. The TLR4 signaling response in COVID-19 may cause influenza to induce acute lung injury/acute respiratory distress syndrome [42] or a TLR4-dependent “cytokine storm” similar to septic shock [43].

Severe SARS-CoV-2 and influenza infections overproduce a large number of proinflammatory cytokines, generating a cascade of events (“cytokine storm”) that is difficult to manage. Severe COVID-19 and influenza infections caused by highly virulent subtypes, such as H1N1 and H5N1, are both characterized by overinduction of the proinflammatory cytokines tumor necrosis factor-α and monocyte chemoattractant protein-1 [44]. In this context, a possible coinfection may induce a positive feedback loop of cytokine production, speeding up the process of respiratory deterioration.

Moreover, both influenza and SARS-CoV-2 have been demonstrated to enhance monocyte tissue factor (TF) expression. TF activates platelets via TLRs that interact with neutrophils to facilitate neutrophil extracellular traps, supporting the process of immunothrombosis [45], ultimately causing (micro)thrombotic complications, such as deep vein thrombosis, pulmonary embolism, and stroke. It is noteworthy that both capillary microthrombi and pulmonary macrothrombi were shown to be significantly more prevalent in patients with COVID-19 than in patients with influenza A [46]. Although no papers studying the dynamic of thrombosis in case of coinfection are yet published, coinfections of influenza with other respiratory pathogens were shown to increase mortality and lead to more severe disease characterized by capillary thrombosis, zones of vasculitis, and necrosis surrounding areas of bronchiolar damage [47].

Additionally, furin is a type 1 membrane-bound protease expressed in multiple tissues that can cleave both SARS-CoV-2 [48] and influenza [49] surface glycoproteins that, in turn, increase viral ability to permeate the host cell membrane. The massive increase of active TGF-β in COVID-19 likely surges furin expression [50], facilitating the binding of ACE2 to the viral S protein through cleavage. Influenza has similar cleavage sites acted upon by furin [49], resulting in higher virulence for both influenza and SARS-CoV-2.

Interestingly, the molecular interplay between SARS-CoV-2 and human lung tissue (from the initial phase of receptor binding to viral replication) was explored in the context of a network analysis in systems biology by constructing lung protein interactomes. Thereby, 50 hub proteins of the replication machinery network were noticed both in influenza and SARS-CoV-2 infection, suggesting similar mechanisms of viral replication and viral-host interactions [51]. This may have important therapeutic implications regarding appropriate treatment strategies in case of coinfection.

A possible protective effect of influenza against SARS-CoV-2 infection was also discussed. The downregulation of ACE2 by the influenza virus [52] may lower susceptibility to infection.

## 5. Effects of Influenza Vaccination on the Risk of SARS-CoV-2 Infection

Since the introduction of influenza vaccination, a significant number of lives have been spared [53]. Each winter, the impact of influenza infection on older people can lead to up to 60,000 deaths in Europe alone [54]. Preceding the influenza pandemic, vaccination of diabetes patients has been shown to reduce hospitalization [55] significantly. Vaccination reduces the risk of ICU admittance of older adults with diabetes (due to respiratory failure and viral pneumonia), significantly lowering the risk of death [56].

Most international management guidelines for patients with chronic respiratory diseases recommend influenza vaccination [57], giving concrete evidence to reduce severe exacerbations and death risk [58]. However, recent data on the effectiveness of the influenza vaccines looking into long-term cell-mediated immunity [59] seems to be a future-proof direction in providing broad protection and optimized vaccination, especially in the abovementioned high-risk groups [60].

The idea of providing cell-mediated immunity is not new, and is based on restoring the Th1 cell immunity while downregulating the cytokine storm with IL-6 [61].

The question (with no valid answer, whatsoever) of using the cell-mediated immunity in a SARS-CoV-2 vaccine has been recently raised [62]. Nevertheless, current data instead sustain the idea that an influenza vaccine can provide cross-immunity protection against coronaviruses infection [63]. It is at least possible that, through the bystander effect [64], the population already vaccinated with influenza vaccine can show less susceptibility or milder symptoms of COVID-19 [65] on top of an obvious protection against the aggressive influenza virus.

## 6. Conclusions

The first conclusion to be retained by practitioners is that a high degree of suspicion regarding coinfection should be maintained. Routine testing protocols will need to be developed for both viruses during the period with an expected high incidence. Data on the severity and clinical course of coinfection are contradictory. Some evidence indicates no worse clinical outcome during coinfection, while others indicate a higher [66], or even lower [67], mortality during the coexistence of COVID-19 and seasonal flu. Due to cross-reactivity between flu and COVID-19, it is advocated that influenza vaccination might have more beneficial effects on COVID-19 than measles, mumps, rubella (MMR) and bacillus Calmette-Guérin (BCG) vaccines [65]. What is clear, for now, is that the more flawed clinical course is due to managing only one of the infections (and subsequently neglecting the other condition).

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
