# Peer review of "The Second Wave of COVID-19 Pandemic Strikes during the Flu Season: An Awareness Perspective"

_medicina, 2020, doi:10.3390/medicina56120707_

Round 1
Reviewer 1 Report
Most parts need a concise and clear introduction.
150 influenza vaccination might have more beneficial effects on COVID-19 than MMR and BCG vaccines[50]. MMR and BCG first appear to need full spelling.
3. Is one plus one equal to two? The clinical outcomes of patients infected with both SARS-CoV-2 and influenza viruses. This part needs more evidence to support the author's opinion.
The title needs a more concise one. A full run of bad luck: the second wave of COVID-19 pandemic strikes precisely during the flu season.
Reviewer 2 Report
This is a timely and interesting review/opinion piece by Berlacu et al. on the importance of co-infections in the ongoing SARS-COV2 pandemic and adds to the body of literature that has been published recently regarding this aspect in the coronavirus infections. The following are some points for the authors to consider to improve the overall strength of the manuscript.
- Section 2: Risk factors for coronavirus infection: Not all the currently known risk factors are included and discussed. For eg, age, heart disease, hypertension, cancer, compromised immunity as in transplantation cases etc are not discussed. Please include all the risk factors while listing category of patients prone to viral infections.
- Section 3: Effect of co-infections: Both viral-viral and bacterial-viral coinfections are mentioned but fugal infections like Chlamydia pneumoniae/Mycoplasma/Aspergillus fumigatus also cause respiratory issues and could be factors for coinfections which is not included in the section, see Kim et al. JAMA 2020 study.
- Section 4: "Cytokine storm" a key feature of immune response associated with Coronavirus infection is mentioned but not discussed in depth. Are there key cytokines/chemokines other than TGFb that could served to distinguish between COVID-19/HINI?
